# Taming Sensitive Weights : Noise Perturbation Fine-tuning for Robust LLM Quantization

Dongwei Wang[1], Huanrui Yang[1*]

[1]Department of Electrical and Computer Engineering, University of Arizona

`dongweiw@arizona.edu, huanruiyang@arizona.edu`

Quantization is a critical step to enable efficient LLM serving under limited resource. However, previous research observes that certain weights in the LLM, known as outliers, are significantly sensitive to quantization noises. Existing quantization methods leave these outliers as floating points or higher precisions to retain performance, posting challenges on the efficient hardware deployment of the mixed-precision model. This work investigates an alternative way to tame the sensitive weights' impact on the quantization error, by reducing the loss Hessian trace with respect to outliers through an efficient fine-tuning process. We propose **N**oise **P**erturbation **F**ine-**t**uning (NPFT), which identifies outlier weights and add random weight perturbations on the outliers as the model going through a PEFT optimization. NPFT tames the sensitivity of outlier weights so that the quantized model performance can be improved without special treatment to the outliers. When applied to OPT and LLaMA models, our NPFT method achieves stable performance improvements for both uniform and non-uniform quantizers, while also offering better inference efficiency. Notably, the simplest RTN can achieve performance on par with GPTQ using our NPFT on LLaMA2-7B-4bits benchmark.

## 1. Introduction

Large Language Models (LLMs), usually with billions of parameters, have demonstrated impressive problem-solving abilities across diverse tasks [1–4]. The enhanced performance, largely driven by the scaling of both training data and model parameters [5], has made it challenging to deploy LLMs on edge computing devices. For example, a model like GPT-3 [6], with 175 billion parameters, requires 350 GB storage space in FP16 and powerful GPUs such as A100 for quick inference, which makes deployment on devices like laptops or mobile phones infeasible without significant model compression. As a promising approach, low-bit weight quantization can help address this issue by enabling efficient inference and reducing storage requirements. Pioneering works such as GPTQ [7] and Squeeze LLM [8] can compress LLaMA [9] model weights to 3-4 bits with a nearly lossless performance, achieving 2-3× speed up on GPUs.

The most straightforward way to perform weight quantization is via linear uniform quantization, which quantizes the entire model to the same bit-width using simple quantizers like Round-to-Nearest (RTN). However, it is widely acknowledged that *weights are not equally important in a neural network*. There is a small fraction of weights that are very sensitive to quantization and lead to significant performance degradation if quantized. This is because these sensitive weights (also referred to as *outliers*) have a larger impact on the outputs of their respective layers, which in turn affects the final loss of the model. As shown in Fig. 1, the performance of two OPT [10] models dropped significantly after applying RTN quantization. However, preserving 0.5% of the outliers as FP16 can greatly recover the performance of the quantized model. A 4-bit quantized model with 0.5% full-precision outliers can achieve performance comparable to that of a full-precision model.

In light of this, existing work typically focuses on reducing the quantization error of outliers by preserving them. For example, SqueezeLLM [8] proposed to store outliers as a sparse FP16 matrix and design a non-uniform sensitivity-based quantizer for other weights. In SpQR [11], outliers are extracted in the form of channel groups and quantized to a higher bit-width. AWQ [12]

Second Conference on Parsimony and Learning (CPAL 2025).

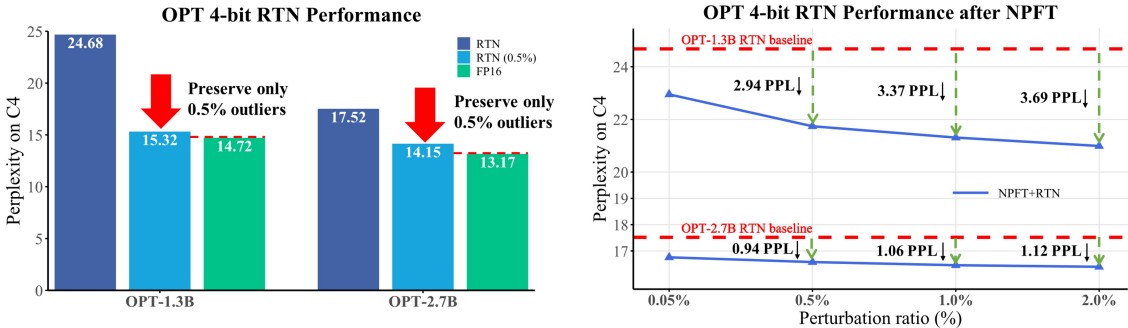

Figure 1: (Left) RTN suffers from the degradation caused by quantizing outliers. Preserving 0.5% of the outliers in FP16 can greatly recover the performance of the quantized model. (Right) NPFT brings significant improvement to the performance of single bit-width models **without preserving any outliers in FP16**. When applied to OPT 1.3B/2.7B models, our method outperforms RTN baseline by a large PPL margin of over **2.9**/**0.9** on the C4 benchmark.

proposed to select outliers based on activation distribution and reduce their quantization error by per-channel scaling. In general, existing works tend to preserve outliers at a higher precision than normal weights. However, even though this may not significantly impact the overall size of the quantized model, it results in the model being in a mixed-precision state. According to [13], most GPUs prefer operations with a uniform data type, as mixed precision requires different storage methods in memory, hindering the full utilization of GPU memory bandwidth. Additionally, mixed precision disrupts the SIMD processing model, leading to inefficient use of GPU cores.

In this paper, we aim to mitigate the special treatment on outlier weights by enabling the model to maintain good performance even after quantizing these outliers to the same bit-width. Drawing inspiration from previous work on analyzing quantization error impacts, like HAWQ [14, 15] and HERO [16], we pinpoint the quantization error of the outliers is contributed by both their quantization errors and the loss function's Hessian trace with respect to them. This observation leads to the proposal of an efficient fine-tuning method named **N**oise **P**erturbation **F**ine-**t**uning (NPFT), which helps reduce the Hessian trace of outlier weights with a short parameter-efficient fine-tuning (PEFT) process. Specifically, we estimate the Hessian trace with the expected model loss under random weight perturbations. We further fine-tune the model to reduce the outlier Hessian trace by performing PEFT optimization with random weight perturbation being added to the outlier weight locations in the base model. The NPFT process avoids the costly higher-order gradient computation needed to optimize Hessian or Fisher matrix directly, while also enjoys a smoother convergence compared to full QAT as all weights are still kept as floating-points in the training process. As shown in Fig. 1, NPFT fine-tuned model shows better PPL than quantized baseline even with the simple RTN quantizer.

In summary, we make the following contribution in this paper:

- We propose NPFT, a PEFT method that efficiently reduces the Hessian trace of outlier weights in LLMs;
- The reduced Hessian trace enables outlier weights to be quantized to the same precision as other weights, without special treatment, while still preserving good quantized model performance;
- NPFT works with different post-training quantizers, enhancing the performance of quantized models across various bit-width settings.

NPFT enables stable perplexity improvements on various uniform and non-uniform quantizers. For example, NPFT helps RTN achieve a 3.69 perplexity improvement on OPT-1.3B-4bits and achieve performance comparable to GPTQ on LLaMA2-7B-4bits. Additionally, by eliminating the need of preserving any outlier weights in FP16, NPFT achieves a 10% reduction in inference latency on the 4090 GPU.

## 2. Related Work

**LLM Quantization.** LLM quantization can be categorized into two branches: quantization-aware training (QAT) methods [17–20] and post-training quantization (PTQ) methods [7, 8, 11, 12, 21]. QAT typically requires extensive retraining to recover accuracy after quantization, whereas PTQ does not involve retraining. Although QAT can enhance the performance of quantized models, it is not easily scalable to LLMs due to the significant computational resources required for retraining and the difficulties in convergence. Therefore, most works on LLM quantization focus on PTQ, which is also the focus of our work. The main difference between our proposed fine-tuning method and QAT lies in our objective: instead of training the quantized weights to recover the performance, we aim to regularize the sensitivity of the weights in the floating point model, making the model more suitable for different PTQ methods. Additionally, our method requires significantly less training time compared to QAT. For example, it only takes one hour of training on LLaMA2-7B.

**Sensitivity-aware PTQ.** There exists a small fraction of weights that are more sensitive to quantization. Quantizing them will lead to significant performance degradation. To address this, sensitivity-aware PTQ methods have been investigated. Some works focus on mitigating outlier activations. For example, [22] preserved outlier activations in floating-point format, while [21] established an outlier suppression framework that transfers outlier factors to other layers. Other works focus on outlier weights, which is also the issue we aim to mitigate in this paper. [11] proposed a hybrid sparse-quantized format where the outlier weights are kept in high precision. [8] also isolated outliers in a sparse FP16 matrix using a Hessian sensitivity-based non-uniform quantizer. [12] extracted the outliers based on activation distribution and performed per-channel scaling to reduce their quantization loss. Even though all these works have achieved promising results, it is important to note that they all place the quantized model in a mixed-precision state, which is unfavorable for hardware deployment. In this work, we aim to reduce the need for special handling of outliers by allowing the model to retain strong performance even when outliers are quantized to the same bit-width.

**Hessian-aware Quantization.** Previous works [14, 15] have shown that the Hessian eigenvalues of the loss function can be used as criteria to determine layer importance in designing mixed-precision quantization schemes. [16] also proved that the model robustness against quantization perturbation can be enhanced by regularizing Hessian eigenvalues. The use of the Hessian to assess the weight sensitivity, or Hessian regularization to improve quantization performance, has been extensively explored in CNN models. However, explicit Hessian regularizations are infeasible for LLMs due to their billions of parameters. In this work, we propose an efficient fine-tuning approach, which can reduce the Hessian trace while bypassing the expensive higher-order gradient calculations required to direct Hessian regularization.

## 3. Method

### 3.1. Identifying Outliers by Hessian Sensitivity

Not all parameters in a neural network contribute equally. In previous works [8, 11], outliers are defined as weights that have a significant impact on the final loss after quantization. Typically, the sensitivity of an arbitrary entry $w_{i,j}$ in weight $W$ can be calculated as the induced loss increase:

$$s_{i,j} = \mathcal{L}(W_q) - \mathcal{L}(W) \tag{1}$$

where $\mathcal{L}$ is the loss function and $W_q$ denotes the weight matrix where $w_{i,j}$ is quantized. We can use Taylor expansion to well approximate the loss increase under quantization as:

$$\mathcal{L}(W_q) - \mathcal{L}(W) \approx g^T(W_q - W) + \frac{1}{2}(W_q - W)^T H(W_q - W) \tag{2}$$

where $g \in \mathbb{R}^{d \times 1}$ and $H \in \mathbb{R}^{d \times d}$ denote the gradient and Hessian of the loss with respect to $W$. $d$ denotes the number of parameters in the weight matrix. $W_q$ and $W$ are flattened into $d \times 1$ vectors in this equation.

For pretrained models that are near convergence, the gradient $g$ approaches zero. The quantization loss will therefore be dominated by the second-order term. In other words, for a fixed quantization bit-width, the sensitivity of weight $w_{i,j}$ is reflected in the corresponding entry in the diagonal of $H$. However, it is infeasible to identify high-sensitivity outliers through direct calculation of $H$, as the computational cost of $H$ for LLMs is prohibitively high. In this paper, we followed [8] and used the Fisher Information Matrix to identify outliers, which can be calculated as:

$$H \approx \mathcal{F} = gg^T = ||\partial \mathcal{L} / \partial w_{i,j}||_2^2 \tag{3}$$

for a specific weight element $w_{i,j}$.

After identifying outliers, previous works typically preserve them in FP16 [8, 11] or quantize them to higher bit-widths [12] to reduce quantization degradation. In this paper, we aim to address this issue at its root by reducing the sensitivity of outliers.

## 3.2. Efficient Hessian Regularization

To mitigate the loss increase brought by quantizing outliers, a straightforward way is to add a regularization term that can minimize the squared sum of the $H$'s eigenvalues [16], thereby reducing the outliers' sensitivity. However, even with the Fisher approximation, directly regularizing $H$ still requires the computation of higher-order gradients, which is computationally unfeasible for LLMs. In this paper, we propose an efficient approach for $H$ regularization.

Fortunately, previous works have provided insights on computing $\text{Tr}(H)$ without direct access to $H$. Based on the fast trace estimation algorithm proposed in [23], $\text{Tr}(H)$ can be estimated using sampled random vector $z \in \mathbb{R}^d$ whose components are i.i.d sampled from a distribution with zero mean and identity covariance matrix. Specifically, the estimation is derived as:

$$\text{Tr}(H) = \text{Tr}(HI) = \text{Tr}(H\mathbb{E}_z[zz^T]) = \mathbb{E}_z[z^T H z] \tag{4}$$

where $I$ is the identity matrix and $\mathbb{E}(\cdot)$ is the expectation.

Considering the random vector $z$ as a perturbation added to the converged weight matrix $W$, the expected loss increase induced by the weight perturbation can be approximated with Taylor expansion, similar to Equation (2), as

$$\mathbb{E}_z[\mathcal{L}(W + z) - \mathcal{L}(W)] \approx \frac{1}{2} \mathbb{E}_z[z^T H z] = \frac{1}{2} \text{Tr}(H), \tag{5}$$

where the first-order term in the Taylor expansion is ignored given the convergence assumption.

As we focus on the weight sensitivity to quantization, we consider a distribution of weight perturbation $z$ that can mimic the impact of quantization on weight values. Specifically, given the quantization bin width as $\Delta$, the round-to-nearest function will change $w_{i,j}$ by at most $\Delta/2$. This suggests that we can represent quantized weight $W_q$ as $W + \delta$, where $\delta \in \mathbb{R}^d, ||\delta||_\infty < \Delta/2$. Therefore, we approximate the distribution of $\delta$ with random weight perturbation $z$ sampled from a zero-mean uniform distribution between $[-\Delta/2, \Delta/2]$. Note that the covariance of $z$ is $\frac{\Delta^2}{12}I$, which is proportional to identity matrix $I$. We can therefore estimate the Hessian trace following the derivation in Equation (4) and (5) as

$$\text{Tr}(H) \propto \mathbb{E}_{z \sim \mathcal{U}[-\Delta/2, \Delta/2]}[\mathcal{L}(W + z) - \mathcal{L}(W)]. \tag{6}$$

Based on the fact that $\text{Tr}(H)$ will be dominated by the outliers of $W$ since their hessian sensitivity is typically more than 100 times greater than that of other weights in terms of scale, it is feasible to reduce outliers sensitivity if we can minimize $\mathbb{E}_z \mathcal{L}(W + z)$ under weight perturbation $z$ applied to the outlier locations. By fine-tuning the model to be robust to weight perturbations on outlier locations, we achieve a computationally-efficient implicit regularization on the Hessian matrix.

## 3.3. Noise Perturbation Fine-tuning

Following the conclusion in section 3.2, the goal of regularizing $H$ is converted into minimize the expected loss under randomly sampled perturbation $z$. Following the stochastic gradient descent

process, we add one independent sample of $z_i$ to the weight in each step of model fine-tuning to fulfill the expected loss computation. Our implementation adopts a two-phase setting as follows.

**Per-channel Noise Sampling.** At the beginning of each epoch, the outlier positions will be identified using the Fisher matrix $\mathcal{F}$ and perturbation ratio $\gamma$. Then $z_i$ will be randomly sampled and added to the corresponding outlier weights in $W$. Previous work [11] has shown that the outliers exhibit strong channel correlations, i.e., outliers are usually concentrated in certain columns of $W$. Therefore, $z_i$ can be sampled per channel and added to all outliers in this channel. The process of sampling and applying noise can be expressed using the following Torch-style pseudocode [1] :

---
**Algorithm 1** Sample and apply noise
---
1: **Input:** weight matrix $W$, Perturbation ratio $\gamma$, Calibration data $X$,
2: **Output:** weight with noise $W + z_i$
3: Calculate $\mathcal{F}$ using $X$
4: $outlier\_positions = \text{Filter}(\mathcal{F}, \gamma)$ # Obtain top $\gamma$% sensitive positions
5: $W_o = \text{torch.zeros}(W.shape)$, $z_i = \text{torch.zeros}(W.shape)$
6: **for** $position$ in $outlier\_positions$ **do**
7:    $W_o[position] = W[position]$ # Obtain outlier weights
8: **end for**
9: $nonzero\_channels = \text{torch.nonzero\_columns}(W_o)$ # Obtain the indices of all channels that contain outliers
10: **for** $col\_idx$ in $nonzero\_channels$ **do**
11:    $noise = \text{torch.rand\_like}(W[:\ col\_idx]) * (W[:\ col\_idx].\max - W[:\ col\_idx].\min) + W[:\ col\_idx].\min$ # Randomly sample noise on this channel
12:    $noise- = noise.\text{mean}$ # Ensure that noise has zero mean
13:    $z_i[:\ col\_idx] = noise$
14: **end for**
15: $W+ = z_i$ # Add noise to $W$
16: **return** $W$
---

**Parameter Efficient Fine-tuning.** For efficient fine-tuning, we apply the LoRA [24]adapters on both self-attention and MLP weights, and merge LoRA weights to the base model after fine-tuning. This pipeline is shown more clearly in Fig. 2. The weight perturbation is added on the corresponding location of the base model weight, so that the gradient computation with respect to the LoRA weight does not require any approximation. To ensure that base model performance is not excessively compromised, we also introduce a weighted $\mathcal{L}(W)$. The overall training objective is:

$$\min_{U,V}[\mathcal{L}(W + z_i + U^T V) + \beta \mathcal{L}(W + U^T V)], \tag{7}$$

where $z_i$ is the weight perturbation following Algorithm 1 and $U, V$ are the LoRA weights.

The proposed fine-tuning process enables the outliers to be less sensitive to the quantization noise, thereby improving the model performance after post-training quantization.

# 4. Evaluations

## 4.1. Experiment Setup

**Models and Datasets.** We perform an extensive evaluation of NPFT across a range of models, including LLaMA [9] and OPT [10] models. We carry out language modeling evaluations using the C4 dataset [25] and the WikiText2 dataset [26]. We also validate NPFT on three common-sense reasoning tasks in Appendix B.

**Baseline Methods.** We compare NPFT against various methods PTQ methods including RTN, GPTQ [7], AWQ [12] and Squeeze LLM [8]. Unless otherwise mentioned, we use Squeeze LLM

---

[1]Note that in Algorithm 1, $W$ and $z_i$ are in matrix form with dimensions $\sqrt{d} \times \sqrt{d}$.

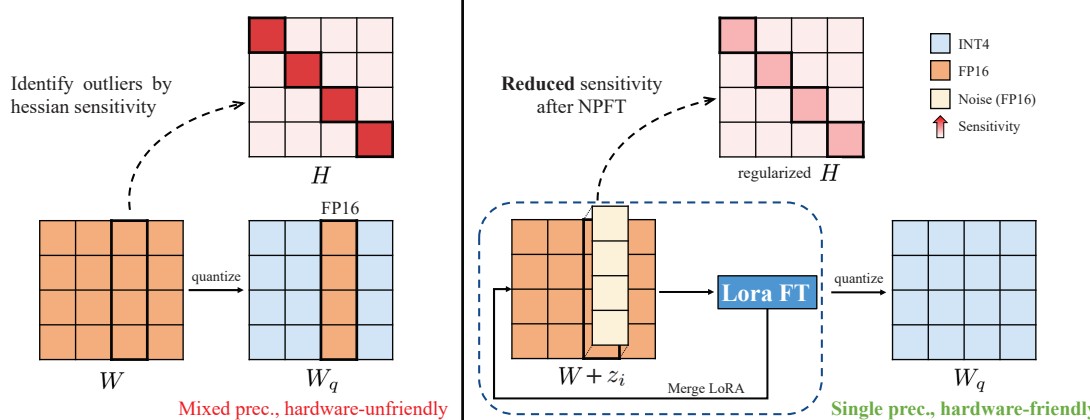

Figure 2: (Left) Existing PTQ methods preserve outliers in FP16 to prevent significant performance degradation, but the mixed-precision format is not hardware-friendly. (Right) Our proposed NPFT regularizes the outliers' sensitivity through efficient fine-tuning, which enhances the performance of single bit-width quantized models without special treatment to the outliers.

without outlier retention as baseline. To conduct fair comparison, we quantize the fine-tuned model using existing quantizers (both uniform and non-uniform) and ensure that the quantizer settings match those in the original works. The results in the experiments are either taken from the original paper or obtained by running its open-source code.

**Efficiency Profiling**. We further compare the latency and memory usage of saving and not saving outliers as FP16 using sqLLM quantizer. Specifically, we measure the latency for generating 128 tokens on a NVIDIA RTX 4090 GPU, using the same optimized cuda kernel in [8]. As NPFT indeed introduces additional training overhead, we also compare the fine-tuning time cost and memory usage of NPFT with QAT method EfficientQAT [17].

**Hyperparameters.** To filter the outliers, we use a calibration dataset comprising 128 randomly selected segments of 512 tokens each from the C4 [25] dataset. Results of different calibration datasets can be found in Appendix C. In the OPT experiments, we identify 0.5% of the sensitive weights as outliers and applied noise to them. The OPT models are fine-tuned for 6 epochs with a learning rate of 5e-6, while in the LLaMA experiments, the perturbation ratio is set to 0.05%, and the model is fine-tuned for 3 epochs with a learning rate of 5e-5. The $\beta$ is set to $0.5$ for all experiments. Noise is applied in an output-channel-wise manner.

## 4.2. Main Results

Tab. 1 shows quantization results for OPT along with representative PTQ methods. Specifically, we use uniform quantizers RTN and GPTQ, as well as the non-uniform quantizer sqLLM to quantize our fine-tuned models and compare PPLs with each quantizer's original results. Note that for sqLLM we only used its dense-only setting, where both normal weights and outliers were quantized. Our NPFT can bring significant improvements to the uniform quantizer. In the OPT-1.3B-4bits experiment, NPFT achieves a 3.69 PPL degradation for RTN on the C4 dataset and 11.89 on WikiText. For GPTQ, NPFT can improve the performance of OPT-2.7B on C4 by approximately 8.5%. SqLLM, as the strongest non-uniform PTQ baseline, can still have its performance further improved by NPFT in the single precision setting. The relatively lower performance enhancement is due to the fact that this quantizer has already reduced the quantization error of outlier weights.

In Tab. 2, we observe that this pattern extends to 7B models. It is worth noting that with the assistance of NPFT, RTN can achieve performance on par with GPTQ for LLaMA-2-7B-4bits. We solely use a small fraction of the C4 dataset as calibration data to filter and fine-tune outlier weights. Meanwhile, NPFT also improves the performance when evaluated on unseen datasets like WikiText.

Table 1: Performance comparison of various quantization methods for OPT-1.3B (top)/2.7B (bottom) on 3-bit and 4-bit settings. Both uniform and non-uniform quantizers are adopted.

| OPT-1.3B | 3-bit | | | 4-bit | | |
|---|---|---|---|---|---|---|
| Method | Avg. Bits | PPL (C4) | PPL (Wiki) | Avg. Bits | PPL (C4) | PPL (Wiki) |
| Baseline | 16 | 14.72 | 14.62 | 16 | 14.72 | 14.62 |
| RTN | 3 | inf. | inf. | 4 | 24.68 | 48.19 |
| RTN + NPFT (0.5%) | 3 | inf. | inf. | 4 | **21.74** | **36.30** |
| GPTQ | 3 | 21.63 | 20.97 | 4 | 16.97 | 15.47 |
| GPTQ + NPFT (0.5%) | 3 | **19.82** | **20.71** | 4 | **15.57** | **15.50** |
| sqLLM | 3.04 | 16.42 | 16.30 | 4.09 | 15.01 | 14.94 |
| sqLLM + NPFT (0.5%) | 3.04 | **16.40** | 16.30 | 4.09 | **14.87** | **14.77** |
| OPT-2.7B | 3-bit | | | 4-bit | | |
| Method | Avg. Bits | PPL (C4) | PPL (Wiki) | Avg. Bits | PPL (C4) | PPL (Wiki) |
| Baseline | 16 | 13.17 | 12.47 | 16 | 13.17 | 12.47 |
| RTN | 3 | inf. | inf. | 4 | 17.52 | 16.92 |
| RTN + NPFT (0.5%) | 3 | inf. | inf. | 4 | **16.58** | **16.32** |
| GPTQ | 3 | 18.17 | 16.88 | 4 | 15.00 | 12.87 |
| GPTQ + NPFT (0.5%) | 3 | **16.59** | **16.85** | 4 | **13.76** | **12.88** |
| sqLLM | 3.04 | 14.45 | 13.85 | 4.07 | 13.38 | 12.80 |
| sqLLM + NPFT (0.5%) | 3.04 | **14.39** | **13.79** | 4.07 | **13.37** | **12.74** |

Table 2: Performance comparison of various quantization methods for LLaMA2-7B on 3-bit and 4-bit settings. Both uniform and non-uniform quantizers are adopted.

| LLaMA2-7B | 3-bit | | | 4-bit | | |
|---|---|---|---|---|---|---|
| Method | Avg. Bits | PPL (C4) | PPL (Wiki) | Avg. Bits | PPL (C4) | PPL (Wiki) |
| Baseline | 16 | 6.97 | 5.47 | 16 | 6.97 | 5.47 |
| RTN | 3 | 404.45 | 542.86 | 4 | 7.72 | 6.12 |
| RTN + NPFT (0.05%) | 3 | **224.39** | **320.64** | 4 | **7.42** | **6.08** |
| GPTQ | 3 | 10.45 | 8.97 | 4 | 7.42 | 5.90 |
| GPTQ + NPFT (0.05%) | 3 | **10.22** | 8.99 | 4 | **7.40** | 5.94 |
| AWQ | 3.24 | 7.84 | 6.24 | 4.24 | 7.15 | 5.72 |
| sqLLM | 3.02 | 7.72 | 6.18 | 4.05 | 7.12 | 5.62 |
| sqLLM + NPFT (0.05%) | 3.02 | **7.69** | **6.12** | 4.05 | **7.08** | **5.60** |

## 4.3. Theoretical Insight Verification

**Reduction of Outliers Sensitivity.** In Fig. 3, we illustrate the changes in outlier sensitivity after applying NPFT. Previous literature has established that the distribution of outliers in the weight matrix tends to exhibit a channel-wise pattern [11]. Therefore, we randomly select an input channel from the weight matrix of OPT-1.3B and unfold it along the output channels to compare the sensitivity of outliers. It is clearly shown that after NPFT, the sensitivity of outliers (both in MLP and self-attention layers) is significantly reduced. This aligns with the conclusions derived in Section 3.2. This also explains why NPFT enhances the performance of quantized models: NPFT helps reduce the performance degradation caused by quantizing outliers.

## 4.4. Efficiency Profiling

**Inference Efficiency.** In Tab. 3, we present the latency and peak GPU memory usage when using the sqLLM quantizer, comparing scenarios with and without retaining full-precision outliers. The outliers are stored using the sparse format described in [8] and utilize the corresponding kernel for inference. When generating 128 tokens on a single 4090 GPU, the uniformly-quantized OPT-1.3B-4bits model trained with NPFT achieves a 10% reduction in latency, lower CUDA memory usage, and demonstrates better performance compared to the model with outliers retained. A similar trend is observed in LLaMA2-7B-4bits model.

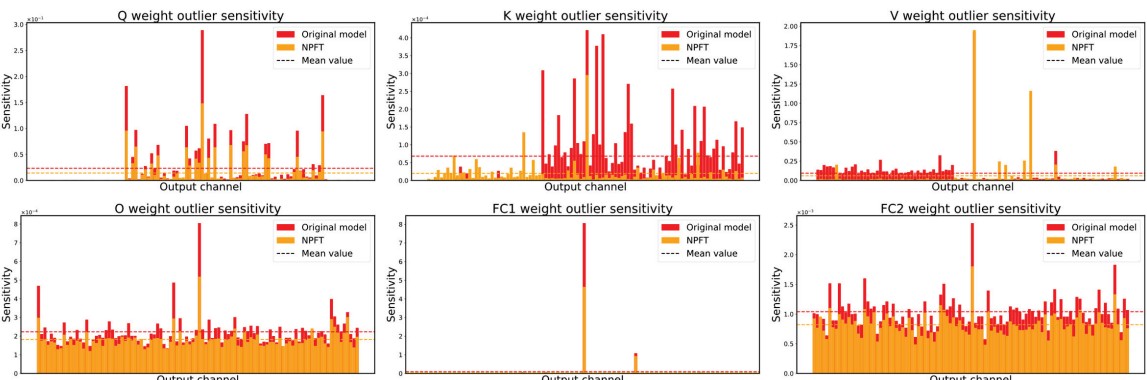

Figure 3: Visualization of outlier sensitivity in OPT-1.3B. The sensitivity is obtained by calculating $\mathcal{F}$. After fine-tuning, the outliers become **less** sensitive compared to the original model.

Table 3: Latency (s) and peak cuda memory usage (GB) of 4-bit models when generating 128 tokens on a 4090 GPU. NPFT enables single-precision models to achieve the same level of perplexity (PPL) as mixed-precision models, while also offering better inference efficiency.

| Method | OPT-1.3B-4bits | | | LLaMA2-7B-4bits | | |
|---|---|---|---|---|---|---|
| | Lat. (s) | cuda Mem. (G) | PPL(C4) | Lat. (s) | cuda Mem. (G) | PPL(C4) |
| sqLLM | 2.40 | 1.08 | 15.01 | 4.80 | 4.46 | 7.12 |
| sqLLM w/0.5% ol. | 2.62 | 1.09 | 14.94 | 5.01 | 4.74 | 7.08 |
| sqLLM + NPFT | **2.40** | **1.08** | **14.87** | **4.80** | **4.46** | **7.08** |

**Training Efficiency.** We also compare the computational overhead with the latest QAT method EfficientQAT [17], whose pipeline also follows a two-phase setting (Block-AP and E2E-QP). As shown in Tab. 4, NPFT's training time for LLaMA2-7B is approximately one-fourth that of EfficientQAT. It is worth noting that, unlike EfficientQAT, which requires dedicated training for each bit-width model, NPFT achieves optimization for multiple bit-width models in one-shot. This not only significantly simplifies the training procedure but also greatly enhances the efficiency of model quantization. NPFT requires significantly fewer samples and shorter sequence lengths for calibration data (shown in Tab. 5), enabling us to perform full-parameter fine-tuning of LLaMA2-7B on a single V100 GPU. The overhead of NPFT mainly comes from memory usage, as the model needs to be loaded into memory when calculating the noise. This memory consumption can be addressed through layer-wise computation in future work.

Table 4: Training time and memory comparison on LLaMA2-7B. NPFT can fine-tune multiple bit-width models in one shot, requiring significantly less training time.

| Method | Phase 1 | | Phase 2 | | Total Time (h) |
|---|---|---|---|---|---|
| | Time (h) | Mem. (G) | Time (h) | Mem. (G) | |
| EfficientQAT (3bits) | 3.3 | 8.5 | 1.5 | 6.4 | 4.8 |
| EfficientQAT (4bits) | 3.3 | 8.5 | 1.5 | 7.0 | 4.8 |
| NPFT (3 & 4bits) (0.05%) | **0.63** | 15.23 | **0.40** | 6.11 | **1.03** |
| NPFT (3 & 4bits) (0.5%) | **0.78** | 15.23 | **0.40** | 6.11 | **1.18** |
| NPFT (3 & 4bits) (1%) | **0.84** | 15.23 | **0.40** | 6.11 | **1.24** |
| NPFT (3 & 4bits) (2%) | **0.91** | 15.23 | **0.40** | 6.11 | **1.31** |

## 4.5. Ablation Study

**Increasing Perturbation Ratio.** In Fig. 4, we show the changes in model performance as the perturbation ratio increases. Note that all groups of the same model are controlled for the same number of training steps. When the perturbation ratio is relatively low (less than 2%), the model's performance does not vary significantly. However, as the ratio increases, it introduces challenges to model

Table 5: Calibration data comparison on LLaMA2-7B. NPFT requires fewer training samples and shorter sequence lengths, thereby reducing hardware requirements.

| Method | Phase 1 | | Phase 2 | | HW Req. | PPL (Wiki) |
|---|---|---|---|---|---|---|
| | samples | Seqlen | samples | Seqlen | | |
| EfficientQAT (3bits) | 4096 | 2048 | 4096 | 4096 | A100 (80G) | **5.81** |
| NPFT w/ sqllm (3bits) | **128** | **512** | **128** | **512** | **V100 (32G)** | 6.12 |
| EfficientQAT (4bits) | 4096 | 2048 | 4096 | 4096 | A100 (80G) | **5.53** |
| NPFT w/ sqllm (4bits) | **128** | **512** | **128** | **512** | **V100 (32G)** | 5.60 |

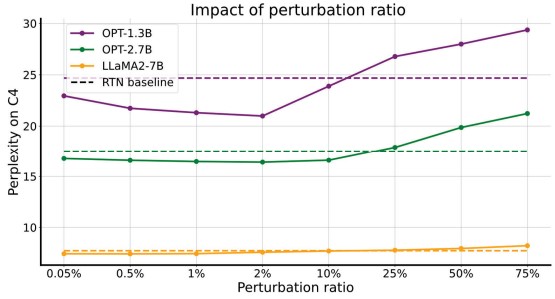

Figure 4: Impact of perturbation ratio on model performance.

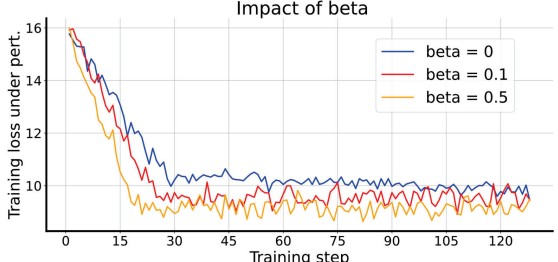

Figure 5: Training loss curves of OPT-1.3B. The perturbed model converges more swiftly with $\beta > 0$.

convergence, indicating a need for longer training times. LLaMA2-7B is more robust to perturbation than OPTs.

**Effectiveness of Induced Base Model Loss.** To retain base model performance, we induce a balanced $\mathcal{L}(W)$ during fine-tuning. As illustrated in Fig. 5, The difference between the loss curves with and without $\beta\mathcal{L}(W)$ suggests that it benefits the model's convergence under perturbation.

**Perturbation on Different Layers.** In Tab. 6, we compare the 4-bit performance changes of the model when perturbing only the attention layers, only the MLP layers, and all layers together. The results show that applying perturbations to all layers yields better performance. Benefiting from LoRA, fine-tuning all layers does not result in a significant increase in training time.

Table 6: Model performance under perturbations applied to different layers. Applying perturbations to all layers yields the best results.

| Model | self_attn | mlp | PPL (C4) | PPL (Wiki) | Training Time (h) |
|---|---|---|---|---|---|
| | ✓ | × | 24.03 | 49.04 | 0.39 |
| OPT-1.3B-4bits | × | ✓ | 23.83 | 37.44 | 0.39 |
| | ✓ | ✓ | **21.74** | **36.30** | 0.40 |
| | ✓ | × | 16.96 | 16.59 | 0.69 |
| OPT-2.7B-4bits | × | ✓ | 16.94 | 16.53 | 0.70 |
| | ✓ | ✓ | **16.58** | **16.32** | 0.72 |

# 5. Conclusion

This work introduces Noise Perturbation Fine-tuning (NPFT), an efficient method to reduce the sensitivity of outlier weights by applying random perturbations during fine-tuning. By reducing the loss Hessian trace, NPFT improves quantized model performance without requiring special treatment for outliers, enhancing both uniform and non-uniform quantizers. Experiments on OPT and LLaMA models demonstrate consistent performance gains and improved inference efficiency. Future work will focus on further optimizing NPFT for larger models and exploring its integration with other quantization techniques to enhance quantization robustness.

# Acknowledgements

This work was supported in part by the research collaboration grant from TetraMem, Inc. This work was based upon High Performance Computing (HPC) resources supported by the University of Arizona TRIF, UITS, and Research, Innovation, and Impact (RII) and maintained by the UArizona Research Technologies department.

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

## A. Performances under different noise distributions and samling methods.

We compare different noise distributions and sampling methods in Tab. 7, with uniform noise and output-channel sampling yielding the best improvements. The noise is effective, regardless of its distribution, as long as it satisfies the zero mean and $Cov \propto I$. The output-channel sampling performs better because outliers are densely concentrated in specific output channels. In contrast, the input-channel method considers only a few weights when computing the scale, leading to a poor approximation of the true quantization.

Table 7: Experimental results of different noise.

| OPT-1.3B | 4-bit | | |
|---|---|---|---|
| Method | Avg. Bits | PPL (C4) | PPL (Wiki) |
| Baseline | 16 | 14.72 | 14.62 |
| RTN | 4 | 24.68 | 48.19 |
| RTN + NPFT (Uniform + o channel) | 4 | **21.74** | **36.30** |
| RTN + NPFT (Gaussian + o channel) | 4 | **22.09** | **35.37** |
| RTN + NPFT (Laplace + o channel) | 4 | **23.18** | **36.80** |
| RTN + NPFT (Uniform + i channel) | 4 | **24.48** | **38.42** |
| OPT-2.7B | 4-bit | | |
| Method | Avg. Bits | PPL (C4) | PPL (Wiki) |
| Baseline | 16 | 13.17 | 12.47 |
| RTN | 4 | 17.52 | 16.92 |
| RTN + NPFT (Uniform + o channel) | 4 | **16.58** | **16.32** |
| RTN + NPFT (Gaussian + o channel) | 4 | **17.04** | **16.60** |
| RTN + NPFT (Laplace + o channel) | 4 | **17.00** | **16.55** |
| RTN + NPFT (Uniform + i channel) | 4 | **17.27** | **17.04** |

## B. NPFT on common-sense reasoning tasks.

We further validated our method on 3 different downstream tasks, including PIQA (Physics), ARC (Science and logic), and Storycloze (Story coherence). The results (accuracy %) are shown in Tab. 8. Consistent improvements are observed on all 3 tasks.

Table 8: OPT models performances on different common-sense reasoning tasks

| OPT-1.3B | 3-bit | | | 4-bit | | |
|---|---|---|---|---|---|---|
| Method | PIQA | ARC | Storycloze | PIQA | ARC | Storycloze |
| Baseline | 72.36 | 50.93 | 70.78 | 72.36 | 50.93 | 70.78 |
| RTN | 52.77 | 27.97 | 47.61 | 67.63 | 49.20 | 59.13 |
| RTN + NPFT (0.5%) | **54.57** | **28.07** | 47.10 | **68.61** | **50.25** | **63.40** |
| GPTQ | 68.34 | 46.17 | 65.25 | 70.73 | 59.97 | 69.64 |
| GPTQ + NPFT (0.5%) | 68.01 | **51.12** | **66.10** | **70.78** | **60.11** | **69.96** |
| OPT-2.7B | 3-bit | | | 4-bit | | |
| Method | PIQA | ARC | Storycloze | PIQA | ARC | Storycloze |
| Baseline | 74.81 | 54.34 | 71.74 | 74.81 | 54.34 | 71.74 |
| RTN | 51.90 | 26.05 | 46.98 | 73.72 | 52.90 | 70.78 |
| RTN + NPFT (0.5%) | **52.72** | **26.60** | 46.91 | **73.77** | **59.09** | **71.16** |
| GPTQ | 71.38 | 48.19 | 68.43 | 73.99 | 53.11 | 70.46 |
| GPTQ + NPFT (0.5%) | **71.57** | **54.12** | 68.11 | 73.66 | **59.55** | **71.23** |

## C. NPFT using various calibration datasets.

We conducted additional experiments with various calibration datasets, including C4, Wiki, and Ptb, as shown in Tab. 9. NPFT consistently improves performance across all datasets. The calibration data in our method identifies outliers, which exhibit generality in language generation tasks, remaining consistent across datasets as the most sensitive weights. These results demonstrate NPFT's robustness in regularizing universal outliers.

Table 9: OPT models performances on different calibration datasets.

| OPT-1.3B | 4-bit | | | |
|---|---|---|---|---|
| Method | Avg. Bits | PPL (C4) | PPL (Wiki) | PPL (Ptb) |
| Baseline | 16 | 14.72 | 14.62 | 20.29 |
| RTN | 4 | 24.68 | 48.19 | 57.30 |
| RTN + NPFT (C4) | 4 | **21.74** | **36.30** | **29.31** |
| RTN + NPFT (Wiki) | 4 | **22.08** | **36.33** | **32.53** |
| RTN + NPFT (Ptb) | 4 | **22.59** | **35.76** | **29.58** |
| OPT-2.7B | 4-bit | | | |
| Method | Avg. Bits | PPL (C4) | PPL (Wiki) | PPL (Ptb) |
| Baseline | 16 | 13.17 | 12.47 | 17.97 |
| RTN | 4 | 17.52 | 16.92 | 31.05 |
| RTN + NPFT (C4) | 4 | **16.58** | **16.32** | **21.21** |
| RTN + NPFT (Wiki) | 4 | **17.04** | **16.58** | **21.63** |
| RTN + NPFT (Ptb) | 4 | **17.20** | **16.80** | **21.89** |

## D. Inference improvements of different context lengths.

As long-context processing is crucial for LLMs, we also added results with varying lengths in Tab. 10. The results show that the latency speedup of our method increases with longer outputs, achieving over a 3s improvement when generating 2048 tokens on RTX4090.

Table 10: OPT-1.3B-4bit latency(s) for generating different lengths of tokens.

| Method | num of tokens latency (s) | | | | |
|---|---|---|---|---|---|
| | 128 | 256 | 512 | 1024 | 2048 |
| Full model | 4.13 | 7.87 | 15.74 | 28.66 | 54.94 |
| sqLLM w/0.5% ol. | 2.62 | 5.21 | 10.47 | 21.15 | 43.65 |
| sqLLM + NPFT | **2.40** | **4.72** | **9.65** | **19.37** | **40.11** |

