# OpenReview forum: "Taming Sensitive Weights : Noise Perturbation Fine-tuning for Robust LLM Quantization"
_CPAL.cc/2025/Proceedings_Track — CPAL 2025 (Proceedings Track) Poster_

### Official Review · Reviewer_ByDR · 2024-12-27
**Review of Submission #18**

**Rating:** 6
**Confidence:** 4

**Review:**

This work investigates the outliers in LLM quantization and mitigates their impact through efficient fine-tuning.

(+) Strengths:
- The paper is well-written and visually appealing, with a clear motivation to address outliers thus avoiding mixed-precision quantization, which often leads to hardware inefficiencies. The proposed method is thoroughly explained with equations and figures.
- Multiple ablation studies are conducted, offering a comprehensive understanding of the effects of different components in the proposed approach.
- Comparisons are provided across various models and baseline methods.

(-) Concerns:
- The main concern is the effectiveness of the proposed method. As shown in Tables 1 and 2, the perplexity improvements are marginal, e.g., 13.38 vs. 13.37 for OPT-2.7B with 4-bit quantization on C4. Evaluating the method on other downstream tasks, such as common-sense reasoning or more challenging generation tasks, could strengthen the impact.
- In Table 3, what is the latency of the full model without quantization? As the reported improvements seem modest (e.g., 2.62 vs. 2.40).
- Can similar improvements be achieved if the weight perturbation is applied to random weights instead of targeting the outliers?

---

### Official Review · Reviewer_Y4R5 · 2025-01-01

**Rating:** 6
**Confidence:** 4

**Review:**

This paper proposes a method called Noise Perturbation Fine-tuning (NPFT) to address the impact of abnormal weights on performance in the quantization of large language models. NPFT introduces noise perturbations for fine-tuning to reduce the sensitivity of abnormal weights to quantization errors, thereby achieving uniform precision quantization, which not only preserves model performance but also improves inference efficiency.

Pros:
1. NPFT reduces the Hessian trace through an efficient fine-tuning process, significantly reducing the performance degradation of abnormal weights after quantization.

2. NPFT can be combined with a variety of quantization methods (such as RTN, GPTQ) and has wide adaptability.

3. This paper provides rich and detailed experimental results, covering multiple models  and different quantization methods


Cons & Questions
1. Although this paper proposes to reduce the sensitivity of abnormal weights through noise perturbation, there is a lack of in-depth analysis of the theoretical impact of noise. For example, how to choose the optimal noise distribution) and the specific impact of noise perturbation on the weight space have not been discussed in detail in theory.

2. The calibration dataset only uses C4, without considering the possible impact of different calibration datasets

3. Although validation was performed on the C4 and WikiText2 datasets, the number of tasks is relatively limited and may not be sufficient to fully evaluate the generalization ability of the proposed method in different tasks and domains.

---

### Official Review · Reviewer_KtoF · 2025-01-13
**An Elegant Solution for Hardware-Friendly LLM Quantization Through Sensitivity Reduction**

**Rating:** 7
**Confidence:** 4

**Review:**

**Strengths**
1. The paper introduces an innovative solution that reduces outlier weight sensitivity through noise perturbation. This enables uniform quantization that is more hardware-friendly while maintaining model performance.
2. The method is remarkably efficient, showing scalability across different model sizes and compatibility with uniform and non-uniform quantizers.

**Weaknesses**
1. The per-channel noise sampling strategy is presented as a design choice, would different noise sampling methods impact the NPFT?
2. Evaluation metrics solely focus on perplexity. Model performance on downstream benchmarks shall be evaluated as well.
3. Template should be updated to CPAL 2025.

---

### Meta-Review · Area_Chair_robp · 2025-02-06

**Recommendation:** Accept (Poster)
**Confidence:** 4

**Metareview:**

This paper provides a novel noise perturbation fine-tuning method for quantization by identifying outlier weights and adding random perturbation to them thereby mitigating their adverse impact on quantization. All reviewers appreciate the novelty of the methodology and empirical results, while noting that theoretical justification is somewhat limited.

---

### Decision · Program_Chairs · 2025-02-11

Accept (Poster)